# Longitudinal field controls vector vortex beams in anisotropic epsilon-near-zero metamaterials

Vittorio Aita [1,3] ✉, Diane J. Roth [1,3] ✉, Anastasiia Zaleska[1], Alexey V. Krasavin[1], Luke H. Nicholls [1], Mykyta Shevchenko [1,2], Francisco J. Rodríguez-Fortuño [1] & Anatoly V. Zayats [1]

Structured light plays an important role in metrology, optical trapping and manipulation, communications, quantum technologies and nonlinear optics. Here, we demonstrate an alternative approach for the manipulation of vector beams carrying longitudinal field components using metamaterials with extreme anisotropy. Implementing vectorial spectroscopy, we show that the propagation of complex beams with inhomogeneous polarization is strongly affected by the interplay of the metamaterial anisotropy with the transverse and longitudinal field structure of the beam. This phenomenon is especially pronounced in the epsilon-near-zero regime, exclusively realised for light polarized along the metamaterial optical axis, strongly influencing the interaction of longitudinal fields with the metamaterial. The requirements on the balance between the transverse and longitudinal fields to maintain a polarization singularity at the beam axis allow control of the beam modal content, filtering diffraction effects and tailoring spatial polarization distributions. The understanding of the interaction of vector beams with metamaterials opens new opportunities for applications in microscopy, information encoding, biochemical sensing and quantum technologies.

The ability to precisely tailor and manipulate the properties of optical wavefronts, such as amplitude, phase, and polarization state, is paramount for the development of various applications including imaging, optical metrology and communications, as well as biochemical and quantum technologies[1–5]. In this respect, complex beam profiling, which exploits the vectorial nature of electromagnetic waves becomes increasingly important[5–9]. Vector vortex beams have emerged as a useful tool for wavefront shaping, thanks to their unique properties offering more degrees of freedom in engineering spatially nonuniform polarization distributions[1,10,11]. These beams have characteristic donut-shaped intensity profiles and can feature polarization singularities with a strong longitudinal (directed along the wavevector) field. Two of the most common types of vector beams with cylindrical symmetry are the so-called radial and azimuthal beams, with radial and azimuthal

orientation of the electric field across the beam profile, respectively. Due to the continuity of the electric field required by Maxwell's equations, radial beams exhibit a strong longitudinal electric field component, centered on the beam optical axis. Conversely, a strong longitudinal magnetic field component together with a purely transverse electric field is found in azimuthal beams. A precise balance between the transverse and longitudinal components of the field is needed in such inhomogeneously polarized beams in order to allow their propagation[12,13].

Typically, longitudinal components are present in guided or strongly focused electromagnetic fields and responsible for their transverse spin[14–16]. This unusual property, which may be present even in unpolarized light[17], is an origin of many counterintuitive effects, such as directional scattering, unidirectional coupling to waveguides,

[1]Department of Physics and London Centre for Nanotechnology, King's College London, London, UK. [2]National Physical Laboratory (NPL), Teddington, Middlesex, UK. [3]These authors contributed equally: Vittorio Aita, Diane J. Roth. ✉e-mail: vittorio.aita@kcl.ac.uk; diane.roth@kcl.ac.uk

lateral optical forces, and photonic topological structures[18–27]. The longitudinal field of vector beams has a profound impact on their scattering properties when they interact with matter. In the case of individual nanoparticles, it induces dipoles in the direction of the beam propagation and, therefore, is scattered in the directions normal to the incident wave vector[28].

Different techniques have been developed for the generation and manipulation of vector vortex beams[29–31], including the use of static phase plates and liquid crystal cells[32–35], as well as spatial light modulators[36,37]. It is also possible to generate vector beams from a non-monochromatic source, exploiting the total internal reflection from a glass cone[38]. Metamaterials and metasurfaces with spatially varying distribution of meta-atoms, which imprint the required phase distribution on an incoming optical field, have also been recently shown to provide an excellent platform for the generation and manipulation of structured beams[39–41], potentially easing their integration into photonic devices.

In this paper, we experimentally and theoretically demonstrate that strongly anisotropic metamaterials–here realized as a two-dimensional array of gold nanorods–can be used to modify the polarization structure and modal content of vector beams, exploiting the drastic difference in the metamaterial response to transverse and longitudinal components of the electric field. When vector vortex beams propagate in such media, the balance between longitudinal and transverse electric field components is broken by anisotropic absorption, requiring the redistribution of the energy between these field components, in order to sustain the propagation. This is especially pronounced in the epsilon-near-zero (ENZ) regime of the metamaterial dispersion for focused radial beams at normal incidence propagating along the optical axis, since the effect on the longitudinal field component is the strongest. Under these conditions, the evolution of the energy distribution between the transverse and longitudinal field components of the vector beam results in the filtering of high-order modes of the beam induced by the focusing and, therefore, diffraction suppression. The understanding of the interaction of vector beams with anisotropic metamaterials opens up new opportunities for applications of polarized beams and nanostructured materials in optical trapping, information encoding, and biochemical sensing.

## Results
### Longitudinal field spectroscopy

Vector vortex beams can be described by the paraxial Helmholtz equation in cylindrical coordinates, seeking a vectorial solution of radial or azimuthal symmetry, respectively[42]. Formally, the solutions obtained are Bessel-Gauss beams, but they are often approximated with the Laguerre-Gauss (LG) modes $LG_{\ell p}$, where $\ell$ and $p$ represent the azimuthal and radial mode numbers. Within the Jones formalism, the transverse field (in the x-y plane normal to the wavevector) of radial and azimuthal vectorial modes can be presented as a superposition of two orthogonally polarized circular vortices[43,44]:

$$\mathbf{E}_{\text{Rad}} = LG_{10}\hat{\mathbf{n}}_R + LG_{-10}\hat{\mathbf{n}}_L \qquad (1a)$$

$$\mathbf{E}_{\text{Azi}} = LG_{10}\hat{\mathbf{n}}_R + e^{i\pi}LG_{-10}\hat{\mathbf{n}}_L , \qquad (1b)$$

where the unit vectors $\hat{\mathbf{n}}_{R,L}$ indicate the right- and left-handed circular polarization states. This representation makes use of two scalar optical vortices $LG_{\pm 10}$ possessing phase singularities of opposite topological charges ($\ell = \pm 1$). The phase of their superposition naturally has a null topological charge, and the center of the intensity distribution represents a singularity in the polarization distribution. The local polarization orientation of radial and azimuthal beams is such that the transverse electric field vector (Eq. (1)) describes a whole $2\pi$ rotation along a closed loop around the beam center, which can define a

winding number, similarly to the topological charge for scalar vortices[45]. Such spatial variation of the transverse polarization enforces a constraint on the longitudinal field that the beam may carry. In the absence of sources, the three-dimensional electric field must be divergence-free, so that the structure of the transverse field requires the presence of a longitudinal field component to satisfy Maxwell's equations[12]: $\nabla \cdot \mathbf{E} = 0$. In particular, the transverse component of an azimuthal beam is sufficient to make it divergence-free, so that this polarization state does not sustain any longitudinal field. On the contrary, a radial beam needs a nonzero longitudinal component $E_{\text{Rad}}^z \neq 0$ to lower its total divergence to zero, even under the paraxial approximation[46]: $\nabla \cdot \mathbf{E} = \nabla_T \cdot \mathbf{E}_T^{\text{Rad}} + ikE_z^{\text{Rad}} = 0$, where $k$ is the wavenumber. This divergence-free condition needs to be satisfied as the beam propagates and interacts with a medium.

To exploit this unique property for controlling vector beams, we used a uniaxial anisotropic metamaterial based on plasmonic nanorod assembly (Fig. 1a, b)[47]. It consists of an array of gold nanorods embedded in an alumina matrix supported by a glass substrate (see Methods for the details of the fabrication). In the absence of pronounced nonlocal effects, the metamaterial optical properties can be described within a local effective medium theory using the Maxwell Garnett approximation[48] (see "Methods"). The metamaterial behaves as a highly anisotropic uniaxial medium with the optical axis parallel to the nanorods and permittivity tensor given by $\boldsymbol{\varepsilon}^{\text{eff}} = \text{diag}[\varepsilon_{xx}, \varepsilon_{yy} = \varepsilon_{xx}, \varepsilon_{zz}]$. For the chosen structural parameters, $\varepsilon_{xx,yy}$ are always positive (Fig. 1c), while $\varepsilon_{zz}$ changes sign at a wavelength around 660 nm, marking the so-called ENZ region that plays a key role in a broad range of applications[47].

The extinction spectra of the metamaterial, measured under p-polarized plane wave illumination, show the typical behavior of this type of nanostructure with two extinction peaks, related to the metamaterial strong anisotropy (Fig. 1d). The one at a wavelength around $\lambda_T \approx 530$ nm is associated with the dipolar resonance (T-mode) excited with a field perpendicular to the nanorod axes. The extinction peak located at a wavelength around $\lambda_{\text{ENZ}} \approx 660$ nm is related to the ENZ behavior of the metamaterial for light polarized along the nanorods and, therefore, observed only at oblique incidence under plane wave illumination. Anisotropic dielectric materials– which do not exhibit an ENZ regime– fail to reproduce the described behavior (Supplementary Fig. S1), demonstrating the fundamental role of the ENZ in the metamaterial sensitivity to a longitudinal field. The effective medium model (see "Methods") describes well the experimental spectra taking into account a correction to the optical properties of the electrodeposited gold for the electron mean-free-path of 8 nm (compared to 10.8 nm of bulk gold)[49–51]. The gold permittivity defines the spectral width of the ENZ extinction resonance (Fig. 1e,f).

Given these properties of the metamaterial, the ENZ-related extinction cannot be harnessed by plane wave illumination at normal incidence, as it does not provide a field component along the optical axis. However, linearly or circularly polarized Gaussian beams can acquire a longitudinal field upon focusing. At the same time, the longitudinal field is intrinsic for radial and other vector vortex beams even within the paraxial approximation[46] and can be further enhanced by focusing.

The choice of the initial state of polarization of the beam before focusing allows for tailoring of the energy exchange between the transverse and longitudinal fields, governed by the requirements of divergence-free electric fields from Maxwell's equations (Fig. 2a). The relative contribution of the longitudinal component to the total intensity generally increases with the numerical aperture (NA) of the focusing. For a radial beam focused in free space, under low focusing conditions (NA = 0.1), the longitudinal intensity corresponds to less than 1% of the overall intensity in the focal plane ($z = 0$), whereas the contribution of the longitudinal field overcomes the one of the transverse field intensity at NA ≈ 0.9. As can be expected, an azimuthal

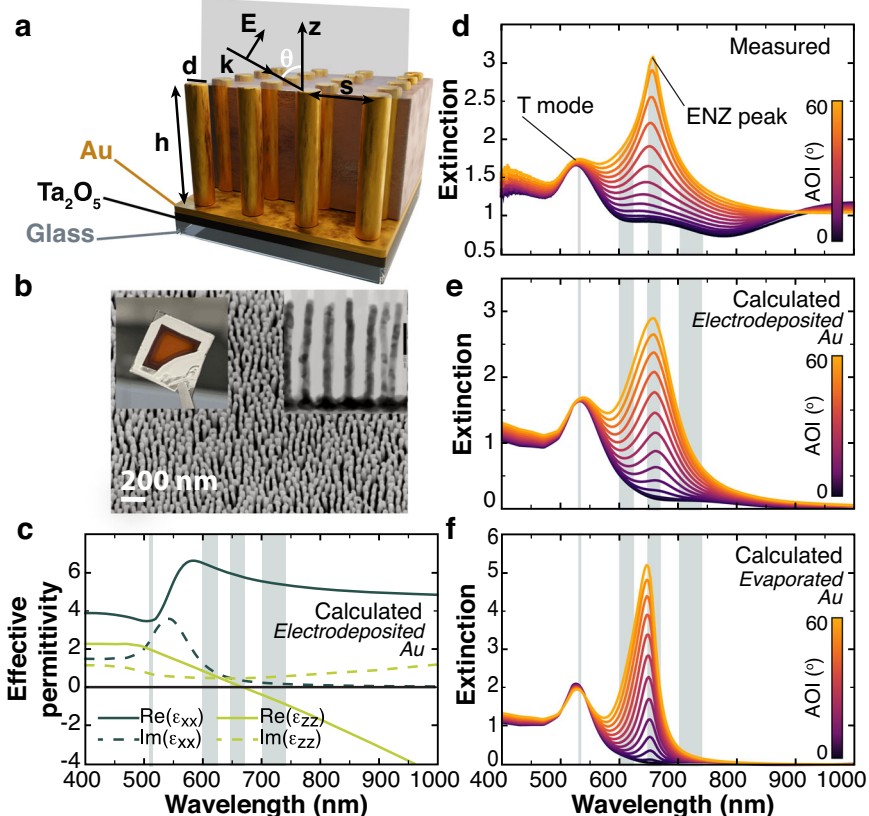

**Fig. 1 | Optical properties of the uniaxial metamaterial. a** Schematics and (**b**) SEM image of the metamaterial, with the insets showing a photograph of the sample and the TEM cross-section of the nanorods. **c** Real and imaginary parts of the effective permittivity tensor for the considered metamaterial ($d = 34$ nm, $h = 400$ nm, s = 69 nm). **d–f** Extinction spectra obtained under *p*-polarized plane wave illumination for increasing angles of incidence, as indicated by the colorbar[67]: (**d**) experiment, (**e**, **f**) effective medium model for the permittivity of (**e**) electrodeposited and (**f**) evaporated gold. Shaded areas in (**c–f**) represent the bandwidth of the filters used for wavelength selection in the experiments for vector field spectroscopy.

beam has a null longitudinal field regardless of the NA, while a linearly polarized beam possesses a nonzero $E_z$ although never as strong as that of the radial beam.

When focused inside the metamaterial, a monochromatic radial beam at a wavelength corresponding to the ENZ condition, acquires a stronger longitudinal field exceeding the transverse field intensity at the much lower NA = 0.4 (Fig. 2b) due to the metamaterial anisotropy, suggesting the possibility to generate a relatively stronger longitudinal component with less focused beams. Similar observations can be made for the linearly polarized beam, while for the azimuthally polarized beam, the longitudinal component keeps the zero value. This effect is strongly wavelength-dependent because of the drastically different interaction between the longitudinal field and the metamaterial in the different dispersion regimes. While the anisotropy increases in the hyperbolic regime compared to the elliptic one, only the ENZ regime produces a strong damping of the longitudinal field[52].

This behavior can be directly visualized in the extinction spectra of the radial and azimuthal beams (Fig. 2). Broadband vector vortex beams can be generated by exploiting a glass cone with a 90° aperture[38]. Double total internal reflection of a circularly polarized beam on the conical surface of the mirror creates a non-uniform state of polarization. Radial and azimuthal beams can be obtained from that state by rotating an additional half-waveplate placed after the cone (see "Methods" for more details). Broadband vector beams generated with the described technique were used to implement vector-beam-based spectroscopy (see "Methods"). The measured spectra reveal a significant extinction of the radial beam at normal incidence in the ENZ regime (at approximately 650 nm), which is related to the interaction of the longitudinal field component of the beam with the metamaterial

ENZ response. The extinction also increases with a stronger focusing responsible for a stronger longitudinal field along the optical axis of the metamaterial, which exhibits the ENZ behavior (Fig. 2c). On the contrary, the azimuthal beam does not carry any longitudinal electric field components irrespective of the focusing. Therefore, azimuthally polarized beams are not affected by the ENZ regime, which influences only the field along the metamaterial axis. This results in the different extinction spectra of radial and azimuthal beams by the metamaterial in the ENZ region (Fig. 2d). Therefore, in the ENZ regime, the metamaterial sensitivity to the angle of incidence shown with plane waves (Fig. 1d–f) can be transformed in a polarization structure sensitivity in the case of strongly focused vector vortex beams. In the experimental extinction spectra obtained with radial beam illumination, the ratio of the magnitudes of the ENZ peak to the T-peak is more than double its theoretical estimate (Fig. 2c,e). Comparing the extinction spectra for the electrodeposited and evaporated gold, one can see that while the widths of the extinction peaks are considerably smaller in the latter case, their magnitudes remain essentially the same, which was not observed for the plane wave case (Fig. 2e, f). These observations could be an indication of the interconnected nature of the transverse and longitudinal fields in the vector beam.

The differences between the propagation in free space and through the metamaterial can be understood as a shift of the balance from transverse to the longitudinal components of the field in the metamaterial case (Supplementary Fig. S2) due to a significant increase of the latter component in the ENZ regime, in accordance with the corresponding boundary conditions at the air-metamaterial interface. The strongly modified field distributions are the result of the transformation of the radial eigenmode over the full vector beam

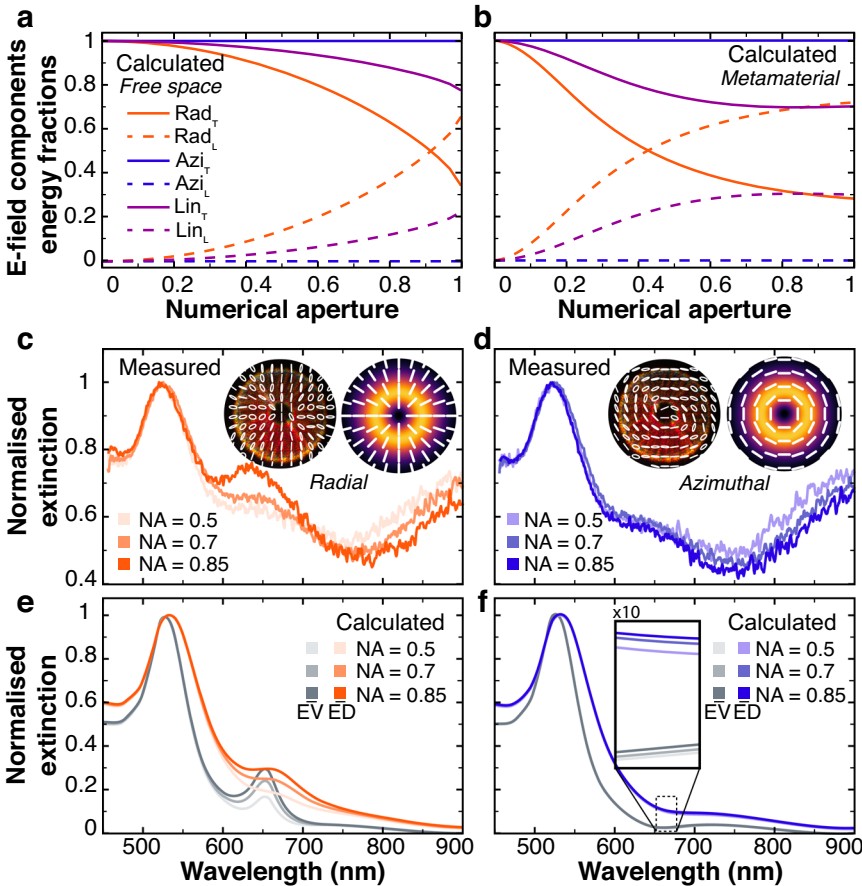

**Fig. 2 | Vector beam spectroscopy. a, b** Relative energy stored in the (solid lines) transverse and (dashed lines) longitudinal components of the electric field with respect to the total energy contained in the beam for propagation in (**a**) free space and (**b**) the metamaterial (the metamaterial parameters are as in Fig. 1c) for (orange) radial, (purple) linear, and (blue) azimuthal polarization states, calculated in the focal plane ($z = 0$) with the metamaterial slab being centered on it.

**c–f** Extinction spectra of the metamaterial obtained with focused vector beams at normal incidence for NA of 0.5, 0.7, and 0.85 (increasing with the color tone): (**c, d**) measured and (**e, f**) calculated extinction spectra of the metamaterial for (**c, e**) radial and (**d, f**) azimuthal beams and for (colored lines) electrodeposited and (greyscale lines) evaporated gold. Extinction spectra are individually normalized to their own maxima for direct comparison.

eigenmode set caused by the presence of the metamaterial layer and the focusing. In the reference case of glass, a strong longitudinal field component is clearly visible at the center of the beam, as well as the presence of higher order modes, as shown by the external rings appearing in the pattern, as a consequence of the focusing/diffraction only. Conversely, the propagation of the beam through the metamaterial differently influences the transverse and longitudinal field components of the beam, depending on the wavelength. The absorption of the longitudinal field component increases as the wavelength gets closer to $\lambda_{ENZ}$, and consequently, the central peak almost completely disappears compared to the non-absorbing case (glass) at the same wavelength. Additionally, the intensity distribution of the transverse field component is structurally modified. Comparing the propagation in glass and in the metamaterial at $\lambda_{ENZ}$, one can observe that the external rings disappear, and the donut-shaped distribution gets smaller in the metamaterial. This indicates that the interaction of focused vector beams with strongly anisotropic medium described above modifies the beam modal content and its polarization state through the tailoring of the longitudinal field, as discussed in the following sections.

### Modal content filtering

Taking into account the observations on the divergence-free nature of the electric field in the absence of sources and the role of focusing in the generation of a strong longitudinal field (Fig. 2), a tightly focused radial beam ensures strong interaction of the longitudinal field with the metamaterial in the ENZ spectral range. At the same time, tight focusing also leads to the appearance of stronger diffraction effects in the propagating beam, resulting in higher-order LG modes contributing to the beam profile. A comparison of the focal plane intensity distributions of a focused radial beam propagating through glass and the metamaterial[52] (see "Methods" for the details of the simulations) shows the strong influence of the metamaterial on the beam profile, which also depends on wavelength (Fig. 3).

The $LG_{\pm1,0}$ modes describing the paraxial radially polarized beam (Eq. (1a)) have the characteristic donut shape of the intensity profiles. Changing the modes quantum numbers affects the mode intensity distribution: higher $|\ell|$ creates wider donuts, $p \neq 0$ causes the appearance of outer rings (Fig. 3c). Importantly, modes $LG_{\pm1p}$ with $p > 0$ carry a strong longitudinal field upon focusing, similar to the radial beam obtained with $p = 0$. Due to the diffraction effects, the intensity distributions in the focal plane can be represented as a superposition of LG modes of higher orders with a nonzero longitudinal field component.

In the case of propagation in glass and in the metamaterial for wavelengths away from the ENZ region, a strong longitudinal peak is present in the center of the focused radial beam (Fig. 3). Close to the ENZ wavelength, two effects can be noticed: both transverse and longitudinal intensity profiles become smoother and the strength of the central peak of the longitudinal field is strongly damped. As a result of the interplay between the modes contained in the field structure and the metamaterial anisotropy, the contribution of higher-order

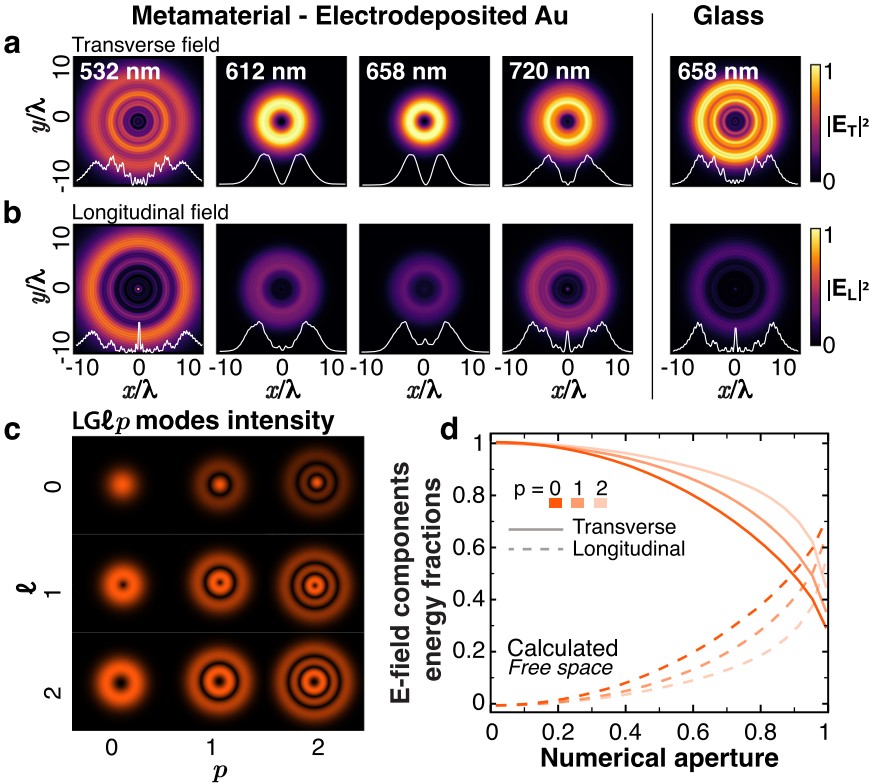

**Fig. 3 | Modal content of a tightly focused radial beam. a, b** Theoretical intensity distribution of (**a**) transverse and (**b**) longitudinal field of a tightly focused (NA = 0.85) radial beam propagating through (four left columns) the metamaterial and (right column) glass for the same wavelengths as in the experiment. Cross-sections of the intensity profiles along the beam axis are shown as the inserts. **c** Intensity distributions of LG modes with $\ell$ = 0–2 and $p$ = 0–2. **d** Relative energy stored in (solid lines) transverse and (dashed lines) longitudinal components of the electric field of radial beams obtained as a superposition of $LG_{\pm1p}$ modes, with $p$ = 0, 1, 2 (orange, decreasing color tone) with respect to their total energy for propagation in free space (cf. Fig. 2a, b).

modes to the electric field is reduced. The absorption of the longitudinal field by the metamaterial requires redistribution of the energy from the transverse field to the longitudinal one in order to maintain the beam propagation, given the relation between these fields prescribed by Maxwell's equations. As a result, the metamaterial effectively filters the higher-order modes out of the beam, leaving the mode $LG_{10}$ dominating over the others.

To confirm the above phenomenological description, the propagation of a focused radial beam and the intensity distributions of its polarization components have been simulated at the wavelength $\lambda_{ENZ}$ for free space, glass, and the metamaterial in order to retrieve the modal content for different focusing conditions (Supplementary Fig. S3). These distributions were then used in a fitting procedure in order to determine the beam modal content (see "Methods"). The experimental results were measured and analyzed using the same procedure.

Both experimental and modeling data confirm that the modal content of a paraxial radial beam predominantly comprises the mode $LG_{10}$ (Fig. 4a, c, d). The modifications of the mode content due to focusing are clearly seen for propagation in free space (Fig. 4c, d), for which the increase in NA shows suppression of a $LG_{10}$ mode in favor of the higher-order modes (mainly $LG_{11}$ and $LG_{12}$). Similar behavior is also observed for propagation through a glass slab with small differences, which can be ascribed to reflection effects (Fig. 4c, d). At the same time, the strong interaction with the metamaterial in the ENZ regime is clearly seen in the modal content of the beam (Fig. 4b–d). The relative contribution of higher-order modes decreases while the $LG_{10}$ mode amplitude increases. This effect is more pronounced for stronger focusing. Experimental results are in good agreement with the theoretical predictions (Fig. 4b).

Hence, high anisotropy and ENZ behavior of the metamaterial can be exploited to efficiently filter higher-order modes generated because

of diffraction and achieve nearly diffractionless focusing of vector beams carrying longitudinal field. Please note that the quality of vector beams is crucial for the experimental observation and applications of both extinction and modal filtering because imperfections of the polarization state (e.g., local ellipticity) can drastically lower the strength of the longitudinal field and the requirements on its presence in the beam profile[12].

## Polarization filtering
In the case of weak focusing, the longitudinal field may not be strong enough to influence the modal structure significantly, however, the beam intensity and vectorial structure may be modified by exploiting the metamaterial anisotropy. The interaction and modification of a paraxial vector beam can be enhanced at oblique incidence when the transverse field component also interacts with the ENZ resonance. In this case, the uniaxial metamaterial can act as a linear polariser or a waveplate, depending on wavelength and angle of incidence[53]. In an isotropic medium (glass), the radial beam experiences only a slight modification of its intensity profile and polarization. It should be noted that the transmission of light through both air-glass and glass-air interfaces favors $p$ polarization over $s$ one because the Fresnel transmission coefficients for the two polarization states are different ($T_p > T_s$) especially for high angles of incidence. Thus, beams with non-uniform polarization states propagating through a glass slide experience a local polarization conversion towards the $p$-polarized state, which leads to the apparent enlargement of the corresponding areas with $S_1 > 0$ in the Stokes parameters maps (Fig. 5). However, for propagation through the metamaterial, the polarization state and intensity profile of the beam are considerably changed and exhibit a strong wavelength dependence. The polarization of the beam is globally converted to linear, while its intensity profile is changed from

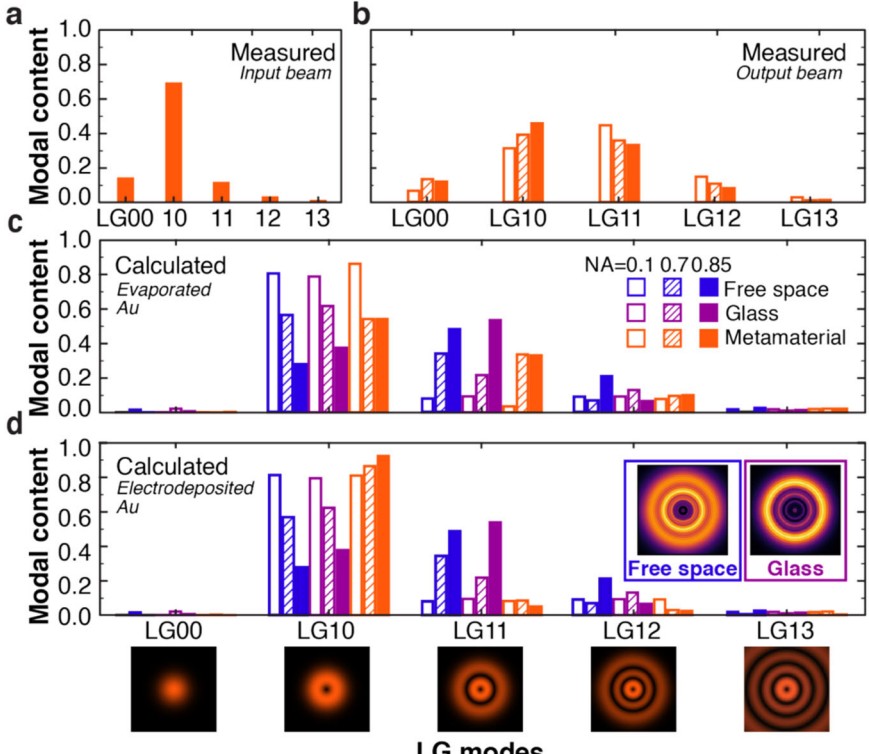

**Fig. 4 | Diffraction suppression of a tightly focused radial beam. a, b** Measured modal content of (**a**) input unfocused (imaged in the *x-y* plane perpendicular to the beam axis at the end of the generation stage, identified in Supplementary Fig. S5) and (**b**) output focused radial beam transmitted through the metamaterial (imaged in the *x-y* plane at the camera location, see Supplementary Fig. S5). **c, d** Simulated modal content of a focused radial beam in (blue) free space, (purple) glass and (orange) the metamaterial for (**c**) evaporated and (**d**) electrodeposited gold

permittivity: (empty bars) NA = 0.1, (patterned) NA = 0.7 and (filled) NA = 0.85. Simulations have been performed in the *x-y* plane located 10 wavelengths from the metamaterial surface. The theoretical spatial intensity distributions of the main LG modes are also shown. Insets in (**d**) show the intensity distributions obtained for a radial beam propagating in free space (blue frame) and glass (purple frame) under tight focusing (NA = 0.85), calculated in the focal plane. Each mode contribution is normalized to the sum of all the mode amplitudes contained in each set.

a donut shape to a two-lobed shape with increasing efficiency while approaching the ENZ region. Simultaneously, the areas of negative $S_1$, which correspond to the regions of vertical polarization in the transmitted beam, become predominant, as the ENZ regime is favouring the transmission of *s* (vertical) polarization components (Fig. 1d). The effect is similar to the transmission of a vector beam through a linear polariser, with the intensity strongly reduced at the points where the original local polarization is parallel to the nanorod axis. The spatial distribution of polarization shows a clear transition from radial and azimuthal polarizations to a linear one. With the reduction of the angle of incidence, the observed effect is gradually reduced until it disappears at normal incidence.

The behavior of the metamaterial as a chromatic linear polariser can be quantitatively characterized by its extinction ratio and bandwidth. The polarization extinction—the ratio of maximum to minimum transmission for two orthogonal linear polarization states—can be obtained from the spectra in Fig. 1. For an angle of incidence of 60°, a polarization extinction ratio of 144:1 is observed with a bandwidth of ~20 nm, centered around $\lambda_{ENZ}$ (Supplementary Fig. S4). The theoretical value corresponds to almost 150000:1 with about 10 nm bandwidth. This illustrates the potential of gold nanorods as a platform for ultra-thin (≤400 nm) wavelength selective polarization devices.

## Discussion

Metamaterials and metasurfaces provide a plethora of opportunities and applications in spatial and/or polarization shaping of light[54–56]. Two-dimensional metasurfaces have become a powerful tool for manipulating complex beams with nonuniform polarizations and can be used both to generate and to manipulate these beams. By designing

metasurfaces to impart spatially varying phase shifts, it is possible to create the helical phase structure of vortex beams or convert linearly polarized light into radially or azimuthally polarized beams. Using cascaded metasurfaces, the purity and control of the generated beams can be improved leading to high-quality vortex beams. These functionalities in beam manipulation, starting from simple polarization filtering to generation and control of optical wavefronts with complex intensity and polarization profiles, are important in numerous applications[57]. Different from two-dimensional metasurfaces, in this work, we experimentally and theoretically investigate the interaction of vector beams with a metamaterial produced by an array of plasmonic nanorods. This metamaterial possesses the properties of a uniaxial crystal, exhibiting extreme anisotropy and ENZ behavior for specific directions of polarization. We showed that with different choices of polarization states and focusing conditions, the longitudinal field of vector beams can be tailored to achieve strong coupling with the ENZ behavior of the metamaterial along its optical axis, even at normal incidence. The anisotropic absorption of the transverse and longitudinal fields and the requirements on the electric field divergence lead to the suppression of the diffraction of a radial beam via manipulation of its longitudinal field. In the paraxial (weak focusing) regime, in which the longitudinal field is weak, the anisotropy of the metamaterial can be taken advantage of at oblique incidence, exploiting anisotropic absorption of the transverse field. Particularly, the intensity and the vector structure of the beam are modified in such a way that the metamaterial acts as a linear polariser with a narrow bandwidth. The spectral position of the anisotropic ENZ regime, which is important for the interaction with a longitudinal field, can be tailored throughout visible and infrared regions at the fabrication stage. Thus,

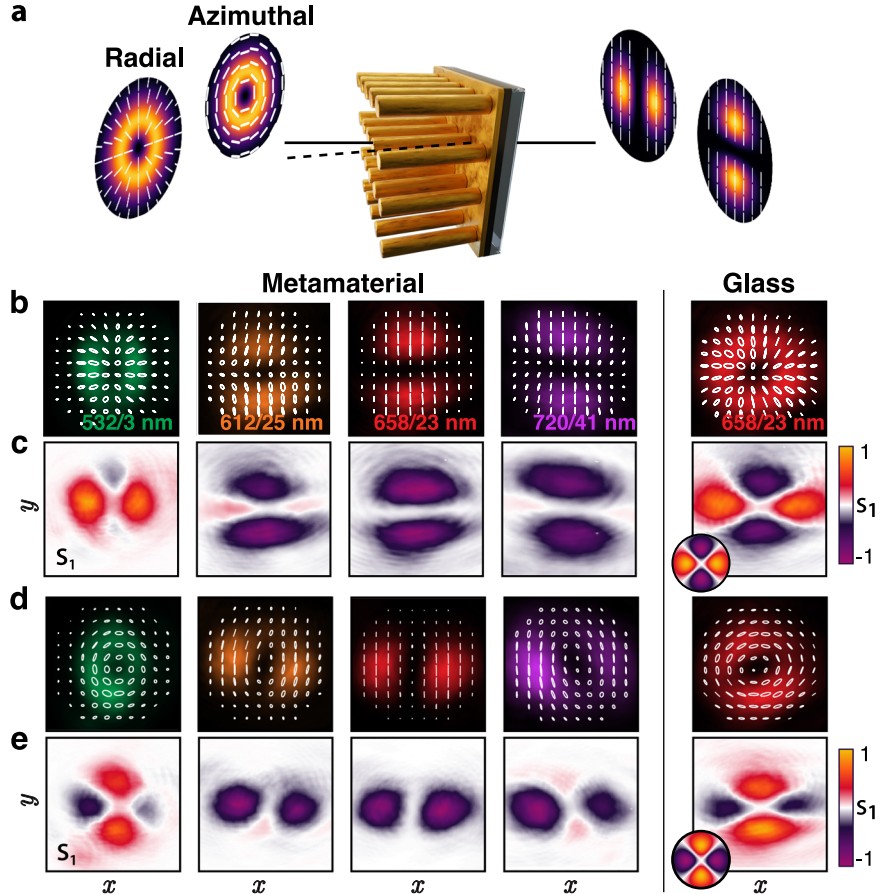

**Fig. 5 | Metamaterial as a linear polariser. a** Schematics of the polarization measurements. **b**, **d** Retrieved polarization of the transmitted light superimposed onto the beam intensity profile and (**c**, **e**) spatial distributions of the Stokes parameter $S_1$ for (**b**, **c**) radial and (**d**, **e**) azimuthal beams, measured in the $x$-$y$ plane at the camera location (see Supplementary Fig. S5). Insets in the last column show the simulated distributions of $S_1$ for propagation in glass at a distance of 10 wavelengths from the surface of a glass slab. Colors of the intensity profiles indicate the illumination wavelength: (green) $532 \pm 1.5$ nm, (orange) $612 \pm 12.5$ nm, (red) $658 \pm 11.5$ nm and (purple) $720 \pm 20.5$ nm. The numerical aperture is NA = 0.1 and the angle of incidence is 60°.

the interaction of vector vortex beams with anisotropic ENZ metamaterials promises the great versatility of such a platform for designed vectorial wavefront shaping, which is of significant interest for various types of applications including optical communications, microscopy, and optical metrology.

## Methods

### Sample fabrication

The nanorod-based metamaterial with controlled geometrical parameters was fabricated by electrochemical deposition of gold into a porous alumina matrix on a supporting substrate[58,59]. The substrate presents a multilayered structure composed of a glass slide (0.7 mm in thickness), a 10-nm-thick tantalum pentoxide ($Ta_2O_5$) adhesive layer and an 8-nm-thick gold film acting as a weakly conducting electrode for the subsequent Au nanorod electrodeposition. An aluminum film (370 nm in thickness) is deposited onto the substrate by magnetron sputtering and subjected to a two-step anodization process in 0.3 M sulfuric acid ($H_2SO_4$) at 25 V to produce a porous anodic aluminum oxide template. The geometrical parameters of the nanopores, such as diameter and separation, are controlled by the anodization conditions (i.e., the choice of the electrolyte acid, its temperature, and anodization voltage) and the duration of pre-deposition chemical etching using a 30 mM sodium hydroxide solution (NaOH). Gold electrodeposition is performed with a three-electrode system using a non-cyanide gold plating solution. The length of the gold nanorods is controlled by the electrodeposition time.

### Effective medium theory modeling

Optical properties of the nanorod-based metamaterial are derived using a local effective medium theory based on the Maxwell Garnett approximation[48], which describes the behavior of a uniaxial crystal. The effective permittivity tensor can then be written in terms of its principal components as $\boldsymbol{\varepsilon}^{\text{eff}} = \text{diag}[\varepsilon_{xx}, \varepsilon_{yy} = \varepsilon_{xx}, \varepsilon_{zz}]$, where the in-plane ($xy$-directions) and out-of-plane ($z$-direction) components of the effective dielectric permittivity are expressed as

$$\varepsilon_{xx} = \varepsilon_{yy} = \varepsilon_{h} \frac{(1+b)\varepsilon_{m} + (1-b)\varepsilon_{h}}{(1-b)\varepsilon_{m} + (1+b)\varepsilon_{h}}, \tag{2a}$$

$$\varepsilon_{zz} = b\,\varepsilon_{m} + (1-b)\varepsilon_{h}, \tag{2b}$$

where $b = \pi(r/s)^2$ is the metal filling factor with $r$ being the radius of the nanorods and $s$ being the periodicity of the array, in the assumption of a square lattice. The permittivities of metal and the host medium are denoted as $\varepsilon_{m}$ and $\varepsilon_{h}$, respectively. This model is valid within the effective medium approximation, i.e., away from the first Brillouin zone edge of the nanorod array. The effective permeability of a gold nanorod metamaterial is unity. The permittivity of the host medium (aluminum oxide, $Al_2O_3$) is taken from ref. 60 and $\varepsilon_{m}$ of evaporated gold from ref. 49. For electrodeposited gold, the permittivity was corrected by setting the restricted electron mean free path to 8 nm[61], which takes into account the variations due to the electrodeposition fabrication process.

The plot of the anisotropic effective permittivity components (Fig. 1d) shows that the metamaterial supports two distinct dispersion regimes: elliptic with $\varepsilon_{xx,yy} > 0$, $\varepsilon_{zz} > 0$ and hyperbolic with $\varepsilon_{xx,yy} > 0$, $\varepsilon_{zz} < 0$. These two regimes are separated by the so-called ENZ region at around a wavelength of $\lambda_{ENZ} = 660$ nm, where $\text{Re}(\varepsilon_{zz}) \approx 0$. From the effective permittivity, the extinction spectra can be evaluated using, for example, the transfer matrix method[62].

## Generation of vector vortex beams

The output beam of a fiber-coupled supercontinuum laser (Fianium, SC-450-2) is expanded to the desired size by a pair of converging lenses in a 2-$f$ configuration (BE) and spatially filtered with a 100 μm pinhole to get a clean beam profile (Supplementary Fig. S5). The initial polarization of the beam is fixed to be horizontal (parallel to the table, $x$ direction) by a Glan-Taylor prism (LP1), after which it is sent to the generation stage of the setup. The setup consists of two branches: the (orange) monochromatic and (purple) broadband generation paths, both coupled to the same detection stage through BS3. The designed beam is brought to the back aperture of an objective (O1) with NA selected among 0.1, 0.5, 0.7, and 0.85, and focused on the sample. Light is collected by a second objective of NA = 0.9 (O2), which is kept the same for each illumination objective. The transmitted beam is sent to a 50–50 beamsplitter cube (BS4) either towards a CCD for image detection and polarization analysis in the case of single wavelength measurements, or coupled to a spectrometer via a multimodal fiber in the case of broadband measurements.

**Broadband generation.** The horizontally polarized broadband light first propagates through a series of double and single Fresnel rhombs, used respectively as half- and quarter-waveplates (HWP2, QWP2) for full control of the scalar polarization of the beam. The beam undergoes a double internal reflection inside a conical mirror of right-angle aperture which causes a spatially varying polarization distribution[38]. The obtained vector vortex beams can be controlled to a certain extent by modifying the input polarization (rotation of HWP2) and rotating the additional double Fresnel rhomb (HWP3) in the output. With a careful orientation of these components, the polarization state can be changed between radial and azimuthal. The orientation of HWP2 and HWP3 can be further adjusted to obtain linearly or circularly polarized light. In this instance, the reflection stage, which consists of a translation stage mounting the cone mirror and a flat circular mirror, is translated in order to introduce a flat mirror in place of the conical one.

**Monochromatic generation.** For monochromatic vortex generation, the chosen wavelength is filtered from the supercontinuum spectrum before the beam is expanded. This is done with colored filters of selected bandwidths and central wavelengths. The monochromatic branch is based on two spatial light modulators (PLUTO2 - NIR011, HOLOEYE), which allow versatility in the type of beams that can be generated[63–65]. Vector vortex beams are obtained as the superposition of scalar vortices of topological charge ±1. Both SLMs are oriented such that their modulation axis is aligned with the $x$-axis, which corresponds to the polarization direction of the source. The first SLM provides a vortex phase term of $\exp(i\ell_1\phi)$, resulting in the Jones vector $|H, \ell_1\rangle$ for the polarization state. This SLM also reflects the beam, giving it an additional $\pi$ phase shift, modifying the beam polarization to $|H, -\ell_1\rangle$. A 4-$f$ telescope is mounted between the two SLMs to reduce the phase variations that would be induced by the propagation between the modulators. Here, a broadband half-waveplate (HWP1), whose axis is oriented at an angle of $\theta_H = \pm\pi/8$ with the horizontal axis, rotates the incoming field by an angle of $\pi/4$, changing its polarization to a diagonal state ($|D\rangle$ or $|A\rangle$). At this stage, the modulation given by SLM1 is stored in both the horizontal and vertical components of the field. When the beam reaches SLM2, its vertical field component is not modulated while the horizontal one acquires a new phase vortex factor

$\exp(i\ell_2\phi)$. Taking also into account the reflection phase shift due to SLM2, the Jones vector after SLM2 becomes

$$|\text{out}_{SLM2}\rangle = \frac{1}{\sqrt{2}} \left( |H, \ell_1 - \ell_2\rangle \pm |V, \ell_1\rangle \right),$$

where the choice in sign depends on whether HWP1 axis angle is set to $+$ or $-\pi/8$. A broadband quarter-waveplate (QWP1) with the axis oriented at $\theta_Q = \pm\pi/4$ with respect to the $x$-direction is then used to transform the $|H\rangle$ and $|V\rangle$ polarization states into $|R\rangle$ and $|L\rangle$, eventually giving as output Jones vector

$$|F\rangle = \frac{1}{\sqrt{2}} \left( |R, \ell_1 - \ell_2\rangle \pm e^{i\Phi}|L, \ell_1\rangle \right).$$

Here, $\Phi$ is the constant phase term representing the phase difference between the two circular vortices that can be tuned by carefully choosing the phase masks on both SLMs. Lastly, a pinhole is placed between L3 and L4 so that the beam is spatially filtered after the modulation procedure has been performed. Using this method, vector vortex beams with an arbitrary topological charge, as well as standard linearly or circularly polarized beams can be generated by a simple change of the phase masks reproduced on the modulators and the waveplates orientation. The choice of the phase masks on SLM1 and SLM2 is dependent on the chosen operational wavelength.

## Vector field spectroscopy

For the extinction measurements under vector vortex beam illumination, the incident beam was defined using the broadband generation technique described above. The beam was directed to the detection branch after transmission through the metamaterial and collected by the spectrometer. To obtain extinction spectra, the average of transmitted signal over a set of 50 repeated measurements, each with the same exposure time of 8 ms was used. The transmitted signal through a glass slab of the same thickness as the substrate of the metamaterial was used as the reference signal.

Extinction measurements for a varied angle of incidence of $p$-polarized plane waves were taken on a similar setup, with the source changed to a high-power halogen lamp and the sample mounted on a rotation motor to allow precise control over the angle of incidence.

## Polarimetry

Polarimetry characterization is performed using monochromatic vector beams, prepared as described above. To obtain the Stokes parameters of the transmitted beam, four images of the beam intensity are taken for LP2 rotated by the angle of 0, $\pi/4$, $\pi/2$, and $3\pi/4$ with respect to the $x$-axis, while two additional images are taken with QWP3 fixed at $\pi/4$ and LP2 either at 0 or $\pi/2$ to create a circular analyzer. With the measured transmitted intensity through the metamaterial denoted with $I(\alpha, \theta)$, where $\alpha$ is the angle of the fast axis of the quarter-waveplate (marked as $\oslash$ in the case when the quarter waveplate is not present) and $\theta$ is the angle of the linear polariser, the Stokes parameters are calculated as

$$S_0 = I\left(\oslash, 0°\right) + I\left(\oslash, 90°\right) \tag{3a}$$

$$S_1 = I\left(\oslash, 0°\right) - I\left(\oslash, 90°\right) \tag{3b}$$

$$S_2 = I\left(\oslash, 45°\right) - I\left(\oslash, 135°\right) \tag{3c}$$

$$S_3 = I\left(0°, 45°\right) - I\left(0°, 135°\right). \tag{3d}$$

To recover the spatial distribution of the polarization states of the transmitted light, additional images of the beam are collected after propagation through LP2. The analyzer is rotated by steps of 20° in the range of $\theta = 0 - 2\pi$. For an electromagnetic wave propagating along the positive $z$-axis with an electric field $\mathbf{E} = (E_x, E_y)$ through a linear polariser with the axis oriented at an angle $\theta$ with respect to the $x$-direction ($\hat{\mathbf{u}} = (\cos\theta, \sin\theta)$), from the measurements of $I(\alpha, \theta)$ at each angle of the polariser, while solving an over-determined system of linear equations, one can fit the three parameters $A$, $B$ and $C$ and subsequently retrieve $|E_x|$, $|E_y|$ and $\cos(\psi_y - \psi_x)$, where $\psi_y - \psi_x$ represents the phase difference between the $x$ and $y$ components of the field. Since one cannot distinguish between left-handed and right-handed elliptically polarized light, in order to retrieve the exact Jones vector, some assumptions on the global phase and handedness of the polarized light need to be made. The global phase factor between $E_x$ and $E_y$ can be chosen such that $\psi_x = 0$, together with the assumption that $\Psi_y > 0$. These assumptions do not change the shape of the polarization ellipse. From the fitting parameters $A$, $B$, and $C$, one can retrieve the polarization components at each point of the beam:

$$E_x = \sqrt{A + B}, \tag{4a}$$

$$E_y = \frac{1}{\sqrt{A+B}} \left[ C + i \sqrt{A^2 - B^2 - C^2} \right]. \tag{4b}$$

### Fitting procedure for modal content analysis

The modal content of the radial beam is retrieved via a custom-made fitting procedure. This uses the same set of images measured (or calculated) for polarimetry description above. The measured (calculated) intensity for the linear polariser axis forming an angle $\theta$ with the $x$-axis can be expressed as

$$I_\theta(r, \phi, z) = |E_x|^2 \cos^2\theta + |E_y|^2 \sin^2\theta + \mathrm{Re}\left(E_x E_y^*\right) \sin 2\theta, \tag{5}$$

where $E_x$ and $E_y$ are the components of the unknown electric field reaching the camera. In principle, each component of this field can be written as a superposition of an infinite number of the LG basis modes. At the same time, the number of present modes can be reasonably limited considering that the size of the modes scales up quickly with their quantum numbers ($\ell$ for the azimuthal angle, $p$ for the radial coordinate). The fitting procedure is based on fixing the maximum values for both quantum numbers ($\ell_{\max}$ and $p_{\max}$) so that a total number of $N = L \cdot P$ modes ($L = \ell_{\max} + 1$, $P = p_{\max} + 1$) is used to decompose the electric field components as follows:

$$E_x(r, \phi, z) = \sum_{\ell, p = 0}^{\ell_{\max}, p_{\max}} A_{\ell p} \, e^{i\varphi_{\ell p}^A} \, \mathrm{LG}_{\ell p}(r, \phi, z) \cos \ell\phi, \tag{6}$$

$$E_y(r, \phi, z) = \sum_{\ell, p = 0}^{\ell_{\max}, p_{\max}} B_{\ell p} \, e^{i\varphi_{\ell p}^B} \, \mathrm{LG}_{\ell p}(r, \phi, z) \sin \ell\phi, \tag{7}$$

where $A_{\ell\mathrm{p}}$, $B_{\ell\mathrm{p}}$, $\varphi_{\ell\mathrm{p}}^A$ and $\varphi_{\ell\mathrm{p}}^B$ are the amplitude and phase of each component, respectively, imposed to be real-valued. The total number of fitting parameters will be $2(N - 1)$, given $\phi^{A_{00}}$, $\phi^{B_{00}} = 0$ so that the fitting procedure only considers relative phases. Examples of the intensity maps fed to the fit, the field resulting from the fit and the main modes contained in it are shown in Supplementary Fig. S3.

### Semi-analytical modeling

The theoretical study of the system is based on a semi-analytical approach described in detail in ref. 52. The approach uses the angular spectrum formalism, based on the Richards-Wolf theory for vectorial diffraction. The developed model can be applied to focused vector beams propagating through multilayered media including layers with anisotropic optical behavior.

## Data availability

All the data supporting the findings of this work are presented in the text and available from the corresponding authors.

## Code availability

All the theoretical findings of this work have been obtained by applying the semi-analytical approach described in ref. 52 with the codes available in ref. 66 as open source.

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

## Acknowledgements

V.A., D.J.R., A.Z., A.V.K., L.H.N., M.S. and A.V.Z. acknowledge acknowledge support from the ERC iCOMM project (789340). V.A., F.J.R.F. and A.V.Z. acknowledge support from the UK EPSRC project EP/Y015673/1. The authors thank Ryo Mizuta Graphics for providing open-access illustration material for optical components used in Supplementary Fig. S5.

## Author contributions

V.A. and D.J.R. contributed equally to this work. V.A., D.J.R., and L.H.N. performed optical measurements; A.Z. fabricated the samples; A.V.K. performed numerical simulations; V.A., M.S., and F.J.R.F. developed the semi-analytical model. A.V.Z. developed the idea and supervised the project. All authors contributed to writing the manuscript.

## Competing interests

The authors declare no competing interests.
