## [Transparent Peer Review file · Nature Communications]

Longitudinal field controls vector vortex beams in anisotropic epsilon-near-zero metamaterials

Corresponding Author: Dr Vittorio Aita

Version 0:

Reviewer comments:

Reviewer #1

(Remarks to the Author)

In this work, the authors propose a new approach for manipulating the vector vortex beams which carry longitudinal field components by using the plasmonic metamaterials with strong anisotropy. It is shown that the propagation of the beams with inhomogeneous polarization is strongly affected by the interplay of the metamaterial anisotropy with polarization structure of the light beam. In the ENZ regime, this kind of phenomenon becomes much more obvious. The proposed strategy may have important applications in wavefront shaping and vectorial polarization engineering. The measured results agree well with the theoretical analysis and well presented. I would like to support the publication of this work after addressing the following comments.

1. For Fig. 2c and 2d, the experimental method for vectorial polarization spectroscopy should be briefly described. Then the link between the main text discussion and Methods section is more clear.
2. In Fig. 5, it seems that the beams with petals are not symmetric, does this come from the laser beam quality or due the inhomogeneity of the nanowires?
3. In the discussion and conclusion part, it might be important to compare the proposed method in this work with the existing technologies (for example, those based on metasurface).

Reviewer #2

(Remarks to the Author)

The manuscript by V. Aita et al. demonstrates the interaction of vector beams with an anisotropic uniaxial metamaterial consisting an array of gold nanorods embedded in an alumina matrix on a glass substrate. The permittivity of the metamaterials approaches zero at some specific wavelengths due to the dispersion relation of the resonant metamaterial. Owing to the intrinsic difference between the longitudinal and transverse electric components of an azimuthal and a radial beam under different NAs, light incorporates various conversion among different polarization during transmission through the anisotropic metamaterial. Specifically, suppression of the diffraction of a radial beam and linear polarization filtering are observed at the epsilon-near-zero region. Overall, the reported phenomena are interesting and are demonstrated by experiments. However, the results are more like phenomenological report, lacking universal physical analysis. Most of the phenomena and effects can be easily imagined or anticipated. For example, the fields or modal distribution must be different during transmission under different NAs owing to the interference effect (in isotropic media) and mode conversion effect (in anisotropic media). Such phenomenon itself is not particularly unusual. Therefore, I would like to recommend it to be considered by other less demanding sister journals at this stage.

1. The unusual thing is the behavior at the epsilon-near-zero region. What is the exact relation between the ENZ dispersion and the suppression of the diffraction or linear polarization filtering? Do they really result from the ENZ property? The reviewer encourages the authors to investigate the underlying relations and physics, which are vital to reevaluate the novelty of the paper.
2. In which plane do they measure the modal contents in Figure 4 and Figure 5?
3. Why the extinction is quite different around 650nm in Figure 2 for radial and azimuthal incident beams?
3. In line 175 on page 4, 'Eq. 2.1' is missing.

Version 1:

Reviewer comments:

Reviewer #1

(Remarks to the Author)

The authors have successfully addressed my previous comments and suggestions, I would like to support the publication of this work in Nature Communications.

Reviewer #2

(Remarks to the Author)

Reply to Reviewers' comments on Manuscript NCOMMS-24-66655

Vittorio Aita, Diane J. Roth, Anastasia Zaleska, Alexey V. Krasavin, Luke H. Nicholls, Mykyta Shevchenko, Francisco J. Rodríguez-Fortuño,
and Anatoly V. Zayats

Reviewer 1

In this work, the authors propose a new approach for manipulating the vector vortex beams which carry longitudinal field components by using the plasmonic metamaterials with strong anisotropy. It is shown that the propagation of the beams with inhomogeneous polarization is strongly affected by the interplay of the metamaterial anisotropy with polarization structure of the light beam. In the ENZ regime, this kind of phenomenon becomes much more obvious. The proposed strategy may have important applications in wavefront shaping and vectorial polarization engineering. The measured results agree well with the theoretical analysis and well presented. I would like to support the publication of this work after addressing the following comments.

Reply: We thank the Reviewer for their high opinion of our manuscript.

[1]. *For Fig. 2c and 2d, the experimental method for vectorial polarization spectroscopy should be briefly described. Then the link between the main text discussion and Methods section is more clear.*

Reply: We thank the Reviewer for the valuable suggestion. In the revised manuscript, we added the required description on p. 8, lines 367–372.

[2]. *In Fig. 5, it seems that the beams with petals are not symmetric, does this come from the laser beam quality or due the inhomogeneity of the nanowires?*

Reply: We thank the Reviewer for highlighting this point. In our opinion, these differences can be ascribed to multiple factors. Firstly, the laser used in the measurements is a fibre-coupled Fianium supercontinuum, whose output intensity distribution is inhomogeneous. While we made all the efforts to clean up the beam, the residual inhomogeneities may make the intensity profile of the beam not perfectly uniform. Another factor contributing to the asymmetric shape is some degree of misalignment of the sample with respect to the objectives used to image it. Light is propagating through either the metamaterial sample or the glass slab at a high angle of incidence (60°) so that an even slight misalignment with the objectives can create astigmatism in the transmitted light. Lastly, there is a physical reason for the lobes shown by S1 to have different brightness. In the case of glass, it is known that s - and p -polarised light is transmitted differently both at the air-glass and the glass-air interfaces, even more so for high angles of incidence. In particular, $T_p > T_s$, as shown in Fig. R1. In the measurement frame of the data shown in Fig. 4 of our manuscript, p - (s -) polarised light corresponds to horizontal (vertical) polarisation. Thus, even in the case of glass, the transmission of horizontally polarised light is favoured against vertically polarised one, leading to the positive petals of S1 being larger than the negative ones. The opposite happens for the metamaterial in the ENZ regime, where it is

Figure R1: **Fresnel coefficients.** Transmission coefficients of a single (a) air-glass and (b) glass-air interface ($n_{\text{air}} = 1$, $n_{\text{glass}} = 1.5$) for s - and p - polarised plane waves incident on the interface at an angle ranging from 0 to 70 degrees. The vertical lines show the angle of incidence in air ($\theta_{\text{air}} = 60^\circ$) and the corresponding angle of incidence in glass ($\theta_{\text{glass}} \approx 35^\circ$).

the p -polarised field component that is favoured in transmission, up to the point that only the negative petals remain. We have added mention of this explanation on lines 619–631 and 638–643.

[3]. *In the discussion and conclusion part, it might be important to compare the proposed method in this work with the existing technologies (for example, those based on metasurface).*

Reply: We thank the Reviewer for the valuable suggestion. We have included a comparison with metasurfaces-based techniques in the discussion section on p. 16, lines 708–728.

Reviewer 2

The manuscript by V. Aita et al. demonstrates the interaction of vector beams with an anisotropic uniaxial metamaterial consisting an array of gold nanorods embedded in an alumina matrix on a glass substrate. The permittivity of the metamaterials approaches zero at some specific wavelengths due to the dispersion relation of the resonant metamaterial. Owing to the intrinsic difference between the longitudinal and transverse electric components of an azimuthal and a radial beam under different NAs, light incorporates various conversion among different polarization during transmission through the anisotropic metamaterial. Specifically, suppression of the diffraction of a radial beam and linear polarization filtering are observed at the epsilon-near-zero region. Overall, the reported phenomena are interesting and are demonstrated by experiments. However, the results are more like phenomenological report, lacking universal physical analysis. Most of the phenomena and effects can be easily imagined or anticipated. For example, the fields or modal distribution must be different during transmission under different NAs owing to the interference effect (in isotropic media) and mode conversion effect (in anisotropic media). Such phenomenon

itself is not particularly unusual. Therefore, I would like to recommend it to be considered by other less demanding sister journals at this stage.

Reply: We thank the Reviewer for these encouraging comments. We would like to point out that while the described phenomena are indeed based on the description of the fields using Maxwell's equations and their consequent effects, and thus in this respect predictable, they have never been demonstrated (and therefore explained) from a point of view of the unique opportunities which artificial optical media with engineered and unconventional optical response bring about. Employing Maxwell's theory, we provide a clear physical explanation of the propagation of cylindrical vector beams in the nanorod metamaterials featuring extreme anisotropy based on the epsilon-near-zero response which we hope will be a basis for their future studies and applications, and, given the strong current interest in these beams, will be an important reading to a broad scientific community. Our work presents not only the experimental observations and numerical modelling but also provides an explanation of the physics based on the properties of the electromagnetic field required to satisfy Maxwell's equations; in particular the behaviour of the longitudinal field of the beams.

[1]. *The unusual thing is the behavior at the epsilon-near-zero region. What is the exact relation between the ENZ dispersion and the suppression of the diffraction or linear polarization filtering? Do they really result from the ENZ property? The reviewer encourages the authors to investigate the underlying relations and physics, which are vital to reevaluate the novelty of the paper.*

Reply: We thank the Reviewer for this important comment and their appreciation of the metamaterial-imposed anisotropic ENZ response and very sorry that our explanation of the physics behind this phenomenon was not clear. We indeed related the unusual interaction of the beam with the metamaterial to the epsilon-near-zero regime and its influence on the beam longitudinal field. We show that the epsilon-near-zero characteristic of the metamaterial is fundamental for the appearance of the phenomena we study in the manuscript. The effects like the suppression of the diffraction and the linearisation of the beam polarisation heavily rely on the interaction between the longitudinal field and the ENZ response of the metamaterial, resulting in the strong absorption of the longitudinal field components. For wavelengths approaching the ENZ regime, the response of the metamaterial to the field polarised along the nanorods is drastically different to the response to the fields orthogonal to the nanorods. Approaching zero, the real part of the zz component of the permittivity results in the extreme discrimination for the vectorial components and hence modal content of the vortex beams. The interaction with the longitudinal field yields the extinction feature appearing at ~ 650 nm shown for a radial beam in Fig. 1(d-f), which is responsible for the effects described in the manuscript. To highlight this, we compared this behaviour to the beam which does not carry a longitudinal field (azimuthally polarised beam), and show in the original manuscript that this absorption feature at the ENZ wavelength is absent. To highlight the fundamental role of the ENZ regime, we have now performed additional simulations of extinction spectra to compare the behaviour of the metamaterial (Fig. R1(a,c)) to a material with equivalent permittivity tensor in all its components but $\Re(\varepsilon_{zz})$, which is instead taken as that of quartz, a typical anisotropic crystal for optical applications. As shown by these calculations, the long-wavelength extinction peak related to the ENZ characteristic of the metamaterial disappears when $\Re(\varepsilon_{zz})$ no longer crosses zero (c.f. Fig. R1(b) and R1(d)). The absence of this effect destroys the metamaterial sensitivity to the longitudinal field component, underlining the fundamental role played by the ENZ response.

We have checked and extended the discussion of the importance of the ENZ behaviour in the revised manuscript (lines 263–268 and 377–390) and added Fig. R1 to the revised Supplementary information as Supplementary Figure 1.

Figure R2: **Role of the epsilon-near-zero regime.** (a,b) Effective permittivity components and (c–f) extinction spectra obtained with the sample illuminated by (c,d) p -polarised plane waves at angles of incidence (AOI) ranging from zero (normal incidence) to 70° or (e,f) a tightly focused radial beam with increasing NA (following the colour tone). The material permittivity represents (a,c,e) the metamaterial studied in this work and (b,d,f) a hypothetical anisotropic crystal with exactly the same permittivity tensor as that of the metamaterial, apart from the real part of its ϵ_{zz} component, which is set to the extraordinary refractive index of quartz, taken from Ref. 1. Following the results shown in Fig. 2 of the original manuscript, panels (e,f) here show a comparison between evaporated (EV) and electrodeposited (ED) gold. In that case, the “quartz-like” sample is obtained starting from either EV or ED gold in the original metamaterial.

[2]. *In which plane do they measure the modal contents in Figure 4 and Figure 5?*

Reply: We thank the Reviewer for the important comment and regret that this was not clear from the text. We have modified the respective figure captions in the revised manuscript to clearly identify the planes, also explicitly identifying the plane positions and the reference system in the revised Fig. 6.

[3]. *Why the extinction is quite different around 650nm in Figure 2 for radial and azimuthal incident beams?*

Reply: We thank the Reviewer for this important question which gives us the opportunity to further clarify the findings described in the manuscript. As shown in Fig. 1(d–f), the metamaterial features an extinction peak at around 650 nm (due to the ENZ behaviour) which is strongly sensitive to the angle of incidence of illumination. Ultimately, this means that the extinction spectrum is sensitive to the presence of a longitudinal field (directed along the metamaterial ENZ axis) and its strength. While under plane wave illumination this component can be achieved by changing the angle of incidence, it is widely known that a longitudinal field is present in a radial beam under any angle of incidence, including normal incidence, as explained in the manuscript on lines 179–233. Maxwell’s equations provide a strict relation between the longitudinal and transverse field components of the beam so that the specific cases of the incident radial and azimuthal beams (shown in Fig. 2) yield two opposite scenarios for comparison. Radial beams have a longitudinal electric field in the centre of the beam (as shown in Fig. 2a) leading to a strong interaction of the beam with the ENZ response of the metamaterial and the related extinction, while the azimuthal beam does not carry a longitudinal electric field component, irrespectively of focusing. Therefore, azimuthally polarised beams are not affected by the epsilon-near-zero regime. This results in the different extinction of radial and azimuthal beams shown by the studied metamaterial in the ENZ regime. Both the experiment and the calculations naturally reproduce this phenomenon. We have checked and specifically modified the description of this phenomenon to make it more clear on lines 377–390.

[1]. *In line 175 on page 4, “Eq. 2.1” is missing.*

Reply: We thank the Reviewer for noticing this. We believe this is part of the numbering system provided by the template for manuscript submission, where Eq. 2.1 stands for Eq. 1 of Section 2. We are very sorry we missed this misprint. We corrected it in the revised manuscript.

References:

[1] Refractive Index database found at <https://refractiveindex.info/>.